# The Role of Technology in Greenhouse Agriculture: Towards a Sustainable Intensification in Campo de Dalías (Almería, Spain)

Antonio J. Mendoza-Fernández [1], Araceli Peña-Fernández [2], Luis Molina [3] and Pedro A. Aguilera [4],*

1 Department of Biology and Geology, Botany Area, CEI·MAR and CECOUAL, University of Almería, Ctra. de Sacramento s/n, 04120 Almería, Spain; amf788@ual.es
2 Research Centre CIAIMBITAL, University of Almería, Ctra. de Sacramento s/n, 04120 Almería, Spain; apfernan@ual.es
3 Water Resources and Environmental Geology, University of Almería, Ctra. de Sacramento s/n, 04120 Almería, Spain; lmolina@ual.es
4 Department of Biology and Geology, Ecology Area, CEI·MAR, University of Almería, Ctra. de Sacramento s/n, 04120 Almería, Spain
* Correspondence: aguilera@ual.es; Tel.: +34-950-015933

**Abstract:** Campo de Dalías, located in southeastern Spain, is the greatest European exponent of greenhouse agriculture. The development of this type of agriculture has led to an exponential economic development of one of the poorest areas of Spain, in a short period of time. Simultaneously, it has brought about a serious alteration of natural resources. This article will study the temporal evolution of changes in land use, and the exploitation of groundwater. Likewise, this study will delve into the technological development in greenhouses (irrigation techniques, new water resources, greenhouse structures or improvement in cultivation techniques) seeking a sustainable intensification of agriculture under plastic. This sustainable intensification also implies the conservation of existing natural areas.

**Keywords:** greenhouse technologies; groundwater; irrigation; land use changes; sustainable agriculture

## 1. Introduction

Agriculture is a key element in the UN Sustainable Development Goals (Objective 2). Two of the aims of this Objective 2 are: To duplicate agriculture production and the income for smallholders for 2030, and to ensure agriculture sustainability. Yet, if the growth of the world's population by 9–10 billion by 2050 is to be taken into consideration together with the fact that this growth will require an increase in food production of between 60–100% [1,2] with an increment of global environmental risks, it may be concluded that there is a certain difficulty in reaching said goals Agriculture's sustainability should not only mean reducing environmental alterations resulting from agriculture practices; but it should also be the strategy for agricultural development and maintenance [3]. In this context, many scientists highlight the importance of agriculture's sustainable intensification [1,2].

Pretty [4] considers agricultural sustainable intensification (ASI) as "significantly increasing production while protecting natural resources". On the other hand, not only does FAO [5] consider the production of more items but also the economic resources together with the reduction of the negative impact on the environment, enhancing natural heritage and ecosystem services flows. This approximation to sustainable intensification implies a holistic view of agricultural sustainable intensification. Currently, sustainable intensification implies more production on the same land area while reducing the environmental impact and maintaining functional ecosystems [6]. ASI is a combined result of various drivers such as social and economic development, policy systems, natural factors, and technological development [7], among which population, smallholders and technology are the most important. Additionally, it is paramount to consider that sustainable intensification

may be diverse and has to be adapted to the location and the context [8]. In this sense, an agricultural system that is intensified: (i) must be adapted to a sustainable intensification considering the landscape scale; (ii) has to take into account the advances in technology on a field scale and its influence at regional/global levels.

As part of Poniente region, in the province of Almería (located in the southeast of Spain), Campo de Dalías is the greatest European exponent of intensive agriculture in greenhouses. The development of this intensive agriculture in a traditionally impoverished region has entailed the development of socio-economic activities, with the rise of living standards for smallholders and the population [9]. Intense land use changes and considerable groundwater exploitation are the main direct pressures that have transformed this region both demographically and economically, as well as in environmental terms [10]. Almeria's intensive agriculture model is characterized by its competitiveness, innovation, and its ability to adapt to market changes or other socioeconomic factors. Currently, the administration and smallholders are exploring new models of sustainability for this intensive system.

The objectives of this paper are twofold: (i) To describe land use changes and groundwater exploitation as the main drivers of transformation; (ii) to make an assessment of the contribution of greenhouse technology for sustainable intensification.

## 2. Materials and Methods

### 2.1. Study Area

Campo de Dalías is located in the southwest of the province of Almeria, in semiarid southeastern Spain (Figure 1). To the region's original municipalities (Dalías, Felix, Vícar and Roquetas de Mar) others of new creation were added (El Ejido and La Mojonera), together with neighboring ones such as Adra, and Berja.

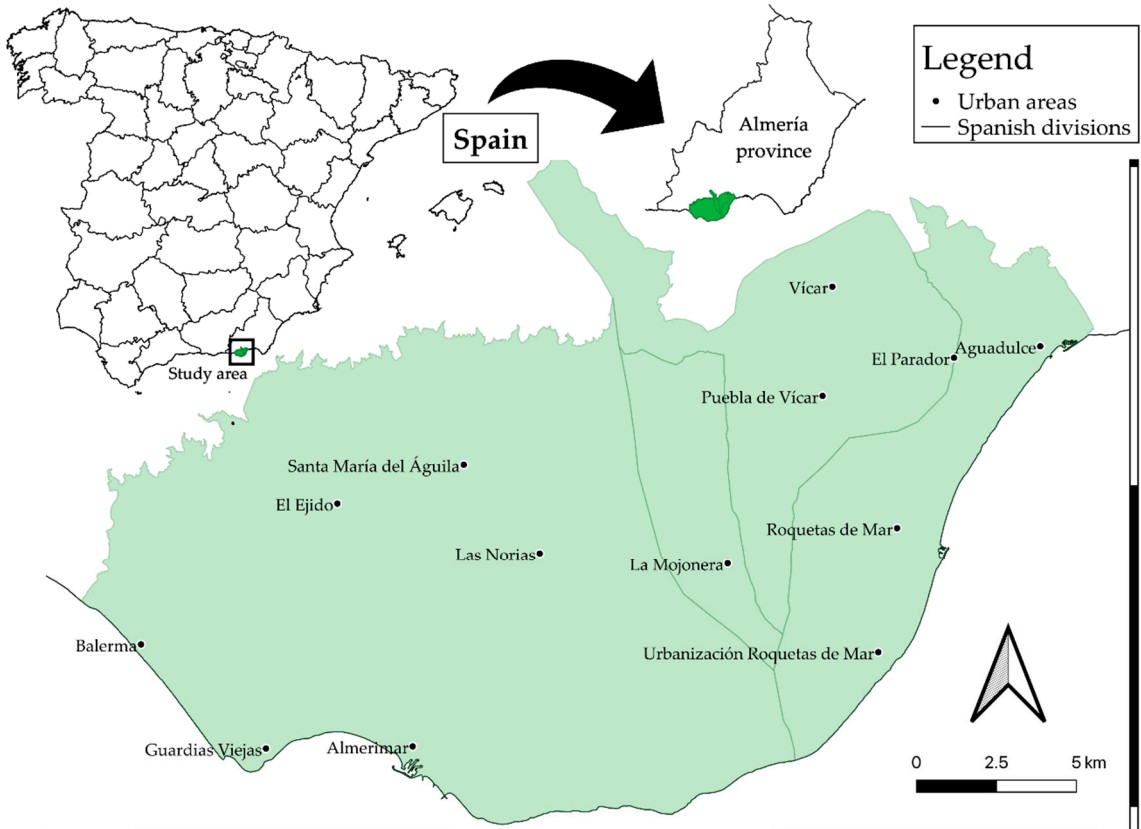

**Figure 1.** Study area. The Campo de Dalías corresponds to the area shaded in green.

From a natural point of view, the area comprises a large, mildly undulating plain of 300 km² that extends from the slopes of Sierra de Gádor to the Mediterranean Sea (Figure 1). Lying on the Alpujáirride basement complex, this mainly flat area descends to sea level in stages that originated from ancient Quarternary sea cliffs [10]. Its relief is a little rugged, sloping gently towards the sea but broken by scarps and a series of tectonic, endorheic basins between Las Norias and El Ejido. The area receives most of its surface water drainage from short, steep watercourses, which flow from the southern face of the Sierra de Gádor and have no outflow to the sea.

The area is characterized by high temperatures, a considerable number of hours of sunshine and scarce rainfall, wind patterns generally from the East or West, and a minimal frequency of frosts. Annual rainfall varies around 200 mm per year. Temperatures are moderate, with the lowest values being recorded in the months of December and January but never below an average of 6 °C. The distribution of rainfall and temperatures is typical of the Mediterranean climate [10–12]. Six types of soils are dominant in the study area. Calcareous regosols, formed from unconsolidated calcium materials in areas with different slopes, which due to their heterogeneity, could be cultivated in dry land. Lithic leptosols, which are characterized by contact with the rock in the first 10 cm of depth, thus significantly limiting tillage. Calcareous cambisols, characterized by a cambic horizon that differs from the underlying horizon by being more or less decarbonated, with a finer texture, or a greater development in their structure. In addition, Fluvisols, Leptosols, and Arenosols may be found in this area [13]. From the agricultural point of view, the soils, in general, are not very fertile, of little depth and high pH. Surface water is scarce. The use of a commodity as limited as water has been a secular problem in the province of Almería. Structures for the collection and storage of rainwater in cisterns are preserved. In the 20th century, an attempt was made to alleviate this problem with the construction of the Benínar reservoir, located outside the region, at approximately 36 km from El Ejido.

Regarding groundwater, three hydrogeological units are differentiated in Campo de Dalías [14,15]. An upper unit, in the center-south, formed by Pliocene calcarenites and quaternary materials (with a thickness of 100–150 m). This unit is separated from two other lower ones by Pliocene marls. The lower units are: Balanegra and Aguadulce, formed by limestone and dolomite from the Triassic age.

*2.2. Mehtodology*

The available maps of uses and land covers of Andalusia were analyzed through a GIS [16] in shape format, having been previously obtained from MUCVA (1956–2007), SIOSE (2011–2013) and SIPNA (2018) projects, which incorporate all the information on land uses and the vegetation cover of Andalusia at 1:10,000 scale [17]. In the study area, the corresponding vector land cover maps for the years 1956, 1977, 1984, 1999, 2003, 2007, 2011, 2013, and 2018 were selected, considering five land-use aggregated classes, namely: Intensive agriculture, scrubland, settlements, traditional agriculture, and water bodies. These years have been used to analyze the changes in Campo de Dalías. Bibliographic information has been used so as to study the evolution of groundwater and greenhouse technology.

**3. Results and Discussion**

*3.1. Changes in Land Use and Groundwater Explotation*

- 1956

At the beginning of the 20th century, the few people who lived in Campo de Dalías dedicated themselves to rainfed agriculture and livestock, both for subsistence, with a small population concentrated in areas of ravines and the central part of Campo. In the 1950s, the Spanish State, through the National Institute of Colonization, began to drill new water wells, and divide the exploitations, building houses and towns for future settlers. The colonization of the territory was carried out with people from other nearby areas around the province of Almería. In the year 1956 (Figure 2), the area hosted a vegetal

community of scrubland (near 27,000 ha), mainly arborescent scrubs with *Ziziphus lotus* (L.) Lam. Human settlements were still scarce and scattered (Figures 2 and 3). Traditional agriculture predominated (with more than 14,000 ha of a mosaic of seasonal crops, namely vegetables and rainfed agriculture) and livestock, with low profitability, mainly including goats and sheep. Regarding water exploitation, in the 1940s, pumpings in Campo de Dalías were around 5–6 hm$^3$, extracted from shallow wells from the surficial aquifers in the western and eastern areas of this territory.

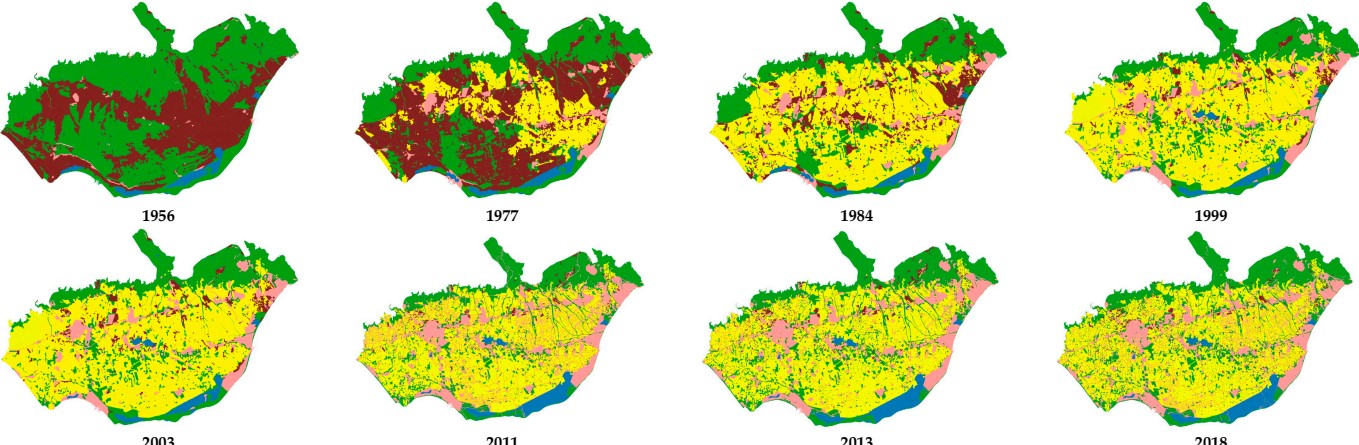

**Figure 2.** Sequential maps of aggregate classes from 1956 to 2018. Intensive agriculture (yellow), scrubland (green), settlements (pink), traditional agriculture (brown), water (blue).

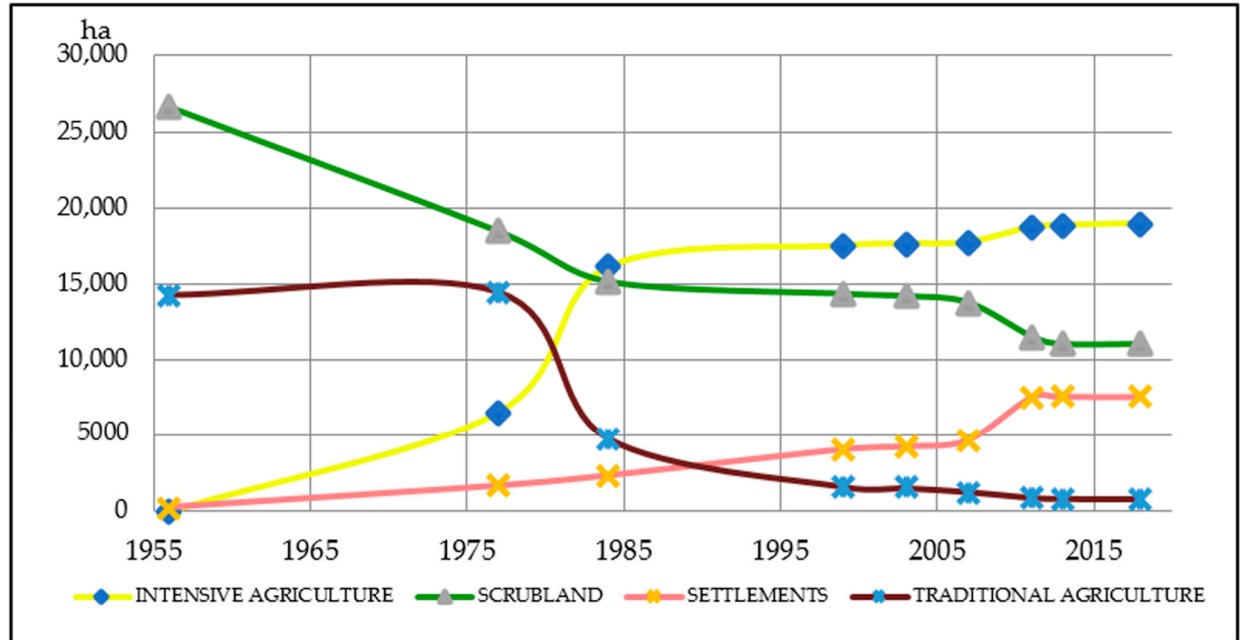

**Figure 3.** Evolution of intensive agriculture, scrubland, settlements, and traditional agriculture from 1956 to 2018.

Later, in the decade of the 1950–1960s, through both private and public initiative, pumping in Campo de Dalías increased considerably, reaching about 30–35 hm$^3$ in the years 1963–1964. In this phase of groundwater extraction, it is considered that the extraction rates were compatible with the replenishment of the aquifers [18]. The crops were carried out in the open air. Due to the strong winds in the area, windbreaks made of canes were incorporated. From 1957 onwards, the sanding technique has been applied to crops [19]. This technique consists in placing a layer of sand about 10 cm thick on top of a layer of manure of 1–2 cm, which is on ravine land. The sand prevents the salts in the water

from rising by capillarity, which allows for the use of low-quality waters, with a high salt content. This reduces the evaporation of the soil by 20–40%. It also increases the earliness of harvests, by accumulating heat in the layer of manure where 50% of the roots develop, improving mobility and absorption of fertilizers.

The first greenhouse was experimentally built in 1963 [20,21] incorporating cultivation in sand. The original greenhouse structures were simple, low, built with wooden posts, wire, and short-term (1 year) polyethylene plastic covering. Their roofs were flat and the plastic sheet had to be perforated to evacuate the rainwater into the greenhouse. In the 1960s, greenhouses represented 1.13% of the total agricultural production in Spain [22].

- 1977–1984

Greenhouse areas in 1977 exceeded 6500 ha. In 1984, the occupied area was over 16,000 ha. Farms were small and mostly family-owned. This increase in the area devoted to intensive agriculture implied a reduction in that of natural scrub by more than 10,000 ha, and a proportional transformation from traditional to intensive agriculture (Figures 2 and 3). The conjunction of several factors such as the low price of agricultural land, the improvement of intensification techniques, and the transport of goods to international markets are the main reasons that the greenhouse area increased so rapidly in the 1980s decade. In this period, demographic movements involved an expansion in the territory dedicated to human settlements, reaching 2320 ha by 1984. In the 1970–1980s, the upper aquifers were mainly exploited. Due to the salinization of the waters, these explorations were abandoned and were replaced by deeper ones, with better water quality. From 1976 to 1984, the exploitation of groundwater almost doubled from 46 $hm^3$ to 99 $hm^3$.

- 1984–2003

In 1990, the area represented 2.48% of the total agricultural production in Spain. Greenhouses occupied a territory of close to 17,500 ha (Figures 2 and 3) in 1999. The natural scrub area was around 14,000 ha. From 1984 to 2003, human settlements almost doubled in size with more than 4000 ha. In contrast, traditional agriculture decreased by more than 3000 ha. The open-air cultivation on sand was relegated to only 200 ha, in 1997 [23]. From a landscape point of view, the homogeneity represented by greenhouses predominated. Due to the expansion of intensive agriculture and the growth of the urban population, the exploitation of groundwater increased in this period, rising from 99 $hm^3$ in 1984 to 123 $hm^3$ in the 2001–2002 period. In 1986, the Spanish Government declared the upper aquifers overexploited [24], forcing the adoption of a series of restrictive measures regarding the expansion of the irrigated area.

Said measures were scarcely applied. The exploitation of the upper aquifer was abandoned due to poor water quality, and the extraction from the lower aquifers began. The abandonment of water withdrawals from the upper aquifer caused the aquifer to gradually recover. In Figure 2, year 1999, two water spots are observed in the center of Campo de Dalías [10]. This wetland is the result of: (i) The extraction of aggregates for agriculture; (ii) the extraction of aggregates below ground level, reaching the water table of the upper aquifer that was in recovery. As a result, a 130 ha wetland was created, called Cañada de las Norias, where endangered species such as *Oxyura leucocephala* (Scopoli) or *Marmaronetta angustirostriss* (Menetries) nest. This wetland has recently been put forward for conservation. In the 1980s, contributions from the Beninar reservoir to the Poniente region began (between 4–10% of surface water).

- 2003–2011

The area of greenhouses did not increase considerably in this interval. However, the horticultural production represented approximately 5% of the total EU production and 21% of that in Spain [25]. Traditional agriculture decreased to about 1000 ha, and the extent of natural scrub areas was also reduced; in 2011, the area covered less than 12,000 ha. Nevertheless, in this period (on 19 July 2006) the SCI (Site of Community Interest) "Artos de El Ejido" was created, in order to ensure the conservation of a representative portion of

this habitat. During this period, the exploitation of groundwater was around 140 hm$^3$. Due to the great agricultural and urban demand for water, aquifers were also overexploited. Most of the greenhouse structures were still made of wood and low in height. In this period, the renovation began using galvanized steel posts and heights over 3 m.

- 2011–2018

In 2018, the greenhouse area covered nearly 19,000 ha. The surface of human settlements exceeded 7500 ha, and in terms of number of inhabitants according to data from the National Institute of Statistics, highlighted Roquetas de Mar (94,925), El Ejido (84,710); Vícar (25,405); Aguadulce (16,176). Traditional agriculture was reduced to minimum values, around 1800 ha. In this time interval, the natural spaces occupied by natural scrubs were also reduced (in 2018 they occupied 11,000 ha). They were located mainly in the southern foothills of Sierra de Gádor. In May 2017, the SCI was proposed by the regional administration to be designated as a Special Area of Conservation (SAC). The biological control of pests and pollination in greenhouses was consolidated, obtaining products without chemical residues. Soil conditioning techniques continued to be applied by adding organic waste products such as manure from different animals, mainly sheep from nearby farms. The overexploitation of the aquifers produced a continuous decrease in piezometric levels in Campo. The current forecast, to support the agricultural and urban demands of the Poniente region, is around 145 hm$^3$/year. A volume of replacement water resources of 100 hm$^3$/year is expected in the coming years, considering that seawater desalination is of vital importance.

At present, the Cañada de las Norias wetland is listed as being of international importance by BirdLife in its catalogue of important areas for birds [26]; it has also been put forward to be listed under the figure of ZEPA, Special Protection Area for Birds. On the other hand, the intensive use of this territory has confined two protected natural spaces: The Special Area of Conservation "SAC Artos de El Ejido" (ES6110014), very deteriorated despite the fact that it represents a priority habitat for conservation in the European Union (5220* Mediterranean arborescent scrub with *Ziziphus*) and unique in the European context [10,27]; and, the Punta Entinas-Punta del Sabinar Natural Area, located in the southern part of Campo de Dalías, with a protected area of 1971 ha, which is home to one of the best-preserved dune systems on the Peninsula. This coastal wetland is fed by drainage waters from the river basin, discharges from the calcarenite aquifer and sedimentary deposits from the basin, and by seawater infiltrations. It has also been catalogued under the protection figures of SAC (ES0000048) and ZEPA since 1989 and since 2006, respectively. The natural area belongs to the Ramsar Convention network of protection.

*3.2. Greenhouse Technologies for Sustainable Intensification*

3.2.1. Evolution of Irrigation Techniques

The use of various technologies, together with greater environmental awareness, have made it possible to reduce water consumption. The techniques focused on limiting water consumption, reducing water loss and on searching for new supplies, acting in the water distribution networks from collection to exploitation and incorporation of desalinated water for irrigation (Table 1). Drip irrigation was introduced in the mid-1970s, which allowed for the volume of irrigation water to be reduced (50%) as compared to furrow flood irrigation that was previously practiced in the area. In 1976, flood irrigation accounted for 97% of total irrigation. In 1998, 100% of crop irrigation was done through drip irrigation [28]. The automation of irrigation to control the application of water and fertilizers to the needs of the plants, fertigation, was one of the greatest technical advances. In 1997, only 22.70% of farmers implemented fertigation. In 2013, 81% of farmers had automatic irrigation programs to control fertigation [29]. With this technique, it is possible to reduce the consumption of fertilizers, increase their efficiency, and reduce contamination by leaching [30]. Fertigation makes it possible to provide a fractionated fertilizer based on the absorption capacity of the crop, bring it closer to the roots, and maintain a low but

constant level of nutrients [31]. The automation of irrigation combined with the use of sensors installed in the ground has especially contributed to saving water: The recorded data can be sent to the computer, telephone, or automatically activate irrigation. The most used soil-moisture sensors are tensiometers, which measure the matrix potential, and are located in the soil close to the roots. Tensiometers indicate the availability of water for the crops, being able to automatically activate the irrigation when the amount falls below the value preset by the farmer, establishing then the irrigation on demand according to the needs of the crop. Tensiometers are cheap, simple, and easy to use, and are suitable for greenhouse crops with high frequency of irrigation and granular soil [32].

**Table 1.** Greenhouse technology with repercussions on the environment.

| Natural Resources | Location | Implemented Technology | | Advantages of Implemented Technology |
|---|---|---|---|---|
| Water | Greenhouse | Drip irrigation and fertigation equipment | | Savings in the consumption of water and fertilizers. Contribution only of crop needs |
| | | Plastic covers in reservoirs | | Prevents evaporation of reservoir water |
| | | Rainwater collection on greenhouse roofs | | Reduction of runoff damage to soils Reduction of water extracted from aquifers Reuse of rainwater for irrigation, reducing the extraction from aquifers |
| | | Introduction of water supply on demand | | Reservoirs are not necessary to guarantee irrigation |
| | Distribution networks | Pipes and channel waterproofing | | Reduction of water losses due to evaporation and filtration |
| | Other sources | Desalinated water | | Reduces extractions from aquifers Improves the quality of water for irrigation Allows other crops not very resistant to salinity |
| Soil | Greenhouse | Sanding | | Reduces evaporation of water from the ground Prevents rising salts by capillarity Increases crop precocity Increased mobility and absorption of fertilizers |
| | | Soilless | | Savings in fertilizers and water. Contribution only of crop needs No risk of aquifers contamination |
| Greenhouse Structures | Greenhouse | Almeria-type greenhouse | | Adapted to the climate of the area Strong, economical structures, with good thermal inertia |
| | | Multi-span greenhouse | | Greater air-tightness and climate control |
| Climate Control | Greenhouse | Passive Techniques (without energy consumption) | Technification of natural ventilation | Temperature and humidity reduction without energy consumption |
| | | | Shading    Whitening Netting | Economical Greater efficiency |
| | | Active Techniques | Cooling system | High cooling efficiency High energy and water consumption |
| | | | Fogging system | Good efficiency with lower water and energy consumption Increase in ambient humidity |
| Cultivation Techniques | Greenhouse | Biological control of pests and pollination | | Elimination of chemical residues in harvested products |



### 3.2.2. Water-Loss Reduction in Distribution Networks and Irrigation Ponds

In 1989, water loss amounted to 19.5% in the distribution networks, and 24% due to an excess of irrigation water in relation to the needs of the crop [33]. Irrigation distribution networks consisted mainly of unlined ditches and concrete pipes, which caused large loss of water through evaporation, seepage, and overflow. In 1997, 59% of the networks were made up of pipes, 29% of lined ditches, which prevented infiltration but not evaporation, and only 12% of ditches on unlined land [33]. According to data provided by the irrigation communities of the Region, currently, almost 100% of the distribution networks are built with closed pipes. Irrigation ponds are used to store water. They are usually made of reinforced concrete with vertical walls that rise above ground level and are rectangular or square. Their depth is usually between 2.50–3.20 m with an average surface of 250–350 m$^2$ depending on the storage needs. Semi-buried or buried ponds with rubber-lined walls were not implanted in the area because they require a larger surface area, which reduced the usable area of cultivation in the plot considerably. Loss due to evaporation of water from the ponds, in the climatic conditions of southern Spain, represents around 15% of the total water supplied to the farms [34]. Covering irrigation ponds with coats results in a decrease in the daily water evaporation rate that reaches 85% when the pond is covered with black double-layer polyethylene [35–37], favoring the condensation process of the water during the night, and therefore, its collection for use [35]. Covers have been gradually incorporated into the irrigation ponds, either by means of shading nets or polyethylene sheets to reduce the evaporation of water. In the climatic conditions of Campo de Dalías, if the irrigation basin is covered and, on the other hand/additionally, the rainwater falling on the greenhouse roof is stored, the water deficit in the basins may be reduced by 53% [38]. From an environmental point of view, uncovered irrigation ponds play an important environmental role in conserving biodiversity [39].

### 3.2.3. New Water Resources

Desalinated water is presented as an alternative to the scarcity of water resources, and an improvement in crop yields compared to currently available water with high salinity [40]. The use of desalinated water from the sea has recently been introduced in the region with the creation of the Campo de Dalías desalination plant (2015), located in the municipality of El Ejido. The capacity of this desalination plant is 30 hm$^3$ per year, of which only 7.50 hm$^3$ are used for irrigation, and it is estimated to cater for 8000 ha of greenhouses; the rest (22.50 hm$^3$) is destined for urban consumption in the municipalities of the area that used to rely on water from the aquifer. The substitution of groundwater for desalinated water will result in the recovery of the aquifers. The commissioning of the desalination plant is presented as an optimal alternative to the scarcity of water resources, and an improvement in crop yields, compared to groundwater with high salinity [40]. Desalinated water has a higher cost than water from a well; it is estimated to reach 5% of the total production cost of the crop, but its use improves productivity and therefore the farmer's net profit increases by up to 25% [41].

### 3.2.4. Greenhouse Structures

Greenhouse structures were developed from wire supporting structures for growing table grapes, a traditional crop in this area. A plastic sheet and an upper mesh were added to the wire mesh, which supported this culture, which was sustained by circular wooden posts. The first greenhouses had flat roofs, were low in height (1.60–2 m), and not very resistant to wind. The enclosure was a polyethylene plastic sheet without additives that was perforated to avoid the storage of water on the roof and caused damage to the crop by dripping, diseases, and excessive environmental humidity. In the 1980s, greenhouses increased in height and resistance, favoring greater thermal inertia; they incorporated roof (zenith) windows and replaced buried stones with concrete pile-like foundations on the perimeter. The wooden columns (props) were replaced by hollow galvanized steel tubes (laminated profiles with a hollow circular section, made of galvanized steel). Cover plastics

with better characteristics were introduced; in 1997, those with low-density polyethylene and 720-gauge thickness were the most used, offering a lifespan of two seasons [29]. In 2013, 800-gauge three-layer plastics (PE-EVA-PE) were used. The incorporation of roof (zenith) windows to the greenhouse structures represented a great advance in the improvement of the interior climatic conditions. In 1997, only 37.70% of the greenhouses had said roof ventilation windows, while in 2013, 98.60% of the greenhouses had already incorporated them. This increase was accompanied by an increase in the rate of ventilation surface. The provision of insect-proof screens in greenhouse vent openings (anti-insect meshes), as a measure of passive protection against diseases, has been obligatory on all windows since 2001. The greenhouse structures in the region correspond to two general typologies: (a) Almería-type greenhouses (that have evolved from the old structures installed in the area; and among which three subtypes can be distinguished depending on the shape of their roof) and (b) multi-span, which is also known as an industrial-type greenhouse (Table 2). In fact, 97.10% of the greenhouses in the region belong to the Almería type. This terminology includes greenhouse structures built with tensioned wire cables and steel or wooden pillars supported on the ground [42]. Depending on the shape of the roof, they are divided into the following: Flat (Figure 4a), asymmetric and gable roof, also called "raspa y amagado" (ridge and furrow) (Figure 4b), having the latter replacing the others (Table 2). They are structures resistant to weather conditions whose average height has gone from 2.80 m (2006) to 4.40 m (2013), multiplying by 2.40 the volume of air inside the greenhouse, also increasing thermal inertia and allowing for the installation of air conditioning elements such as thermal screens, shade nets, or hot-air generators (wind turbines). The predominant roof angle is 13°, twice that of 2006, thus improving solar capture. The industrial greenhouse, with its multi-span (multitunnel) variants ending in a semi-cylindrical and pointed roof (Gothic type), represents slightly less than 3% of the total existing greenhouse area in the Region (Table 2). From a technical point of view, it is more water and airtight than the Almería type, which allows for better control of the climate inside. It is more diaphanous due to the greater distance between the props of the resistant portal frame (gantry) (5–9 m), and a greater separation between portal frames (distance between portal frames gantries (4–5 m), which favors the mechanization of the crop and the installation of auxiliary elements for climate control. They are more resistant structures, but also more expensive.

**Table 2.** Evolution of typology (% of each type with respect to the total) of greenhouses in the Poniente region. Authors' elaboration from source [43].

| Greenhouse Typology | Roof Shape | 1999 | 2006 | 2014 |
|---|---|---|---|---|
| Almeria Type | Flat-top. Ridge and Furrow | 64.20% | 32.20% | 15.20% |
| | Asymmetric | 3.50% | 2.40% | 6.10% |
| | "Raspa y amagado" | 29.20% | 62.50% | 75.80% |
| Industrial Type | Gabled roof multi-span | 2.70% | 0% | 0% |
| | Semi-cylinder multi-span | 0.40% | 2.50% | 1.50% |
| | Gothic multi-span | 0% | 0% | 0.80% |

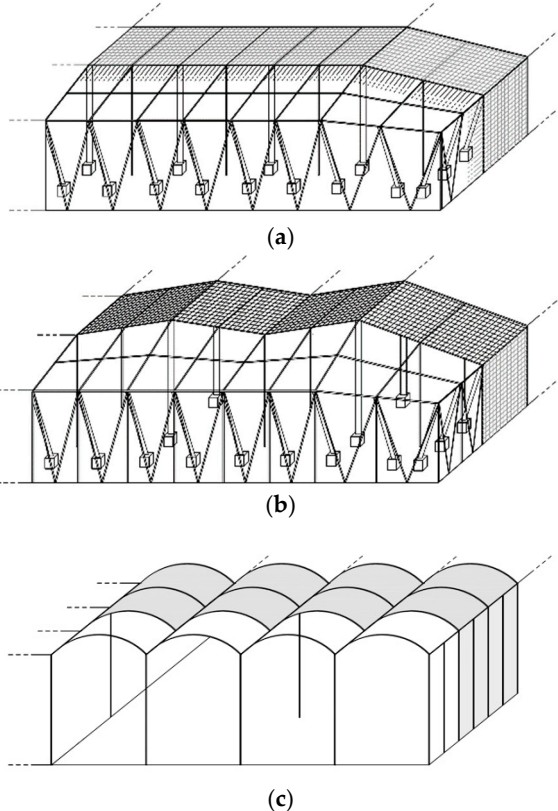

**Figure 4.** Greenhouse typology: (**a**) Flat-top greenhouse, (**b**) Almeria-type "raspa y amagado" greenhouse with gable roof, (**c**) semicylinder multi-span film-covered greenhouse.

### 3.2.5. Indoor Greenhouse Climate Control Techniques

Due to the climatic characteristics of the area, mild winters, and hot summers, various techniques are used to control the interior temperature of the greenhouses. One hundred percent of them incorporate passive shading and ventilation: These do not consume energy, reduce excess radiation during the warmer months, and help cooling. The most widely used shading technique is to paint the exterior walls and the roof white. The effect of this technique depends on the amount of paint applied [44]. The use of reflective shade screens installed inside the greenhouse is not as widespread (reflective shade screens), and is restricted to multi-span greenhouses. Active cooling techniques include the evaporative cooling system, using evaporative pads and extractor fans, together with the fogging system, whose use has not spread due to the high energy and water consumption. The reduction in temperature and water consumption of a fan-pad system installed in a greenhouse in Almería has already been studied. Franco et al. [45] emphasize the importance of regulating the speed of the air entering the greenhouse, depending on the temperature requirements, thus minimizing energy and water consumption. In one greenhouse in Almería, despite the high relative humidity of the air in the hottest hours of the day, a decrease of almost 6 °C in the mean temperature and a water consumption of 13.55 l/h per square meter of an evaporative cooling pad are achieved [46], whose efficiency decreases over time as a consequence of salt encrustations. Furthermore, the study of the natural ventilation in a multi-span greenhouse with one roof and two side vents by means of sonic anemometry has proven that opening the roof vent windward, one side vent leeward, and the other side vents windward causes opposing thermal and wind effects [47].

Given the area´s climatic conditions, the use of heating systems amounts to 8.40% in the greenhouses of Almeria. The installation of heating systems is closely linked to the structure type. These are implemented in 67% of multi-tunnel type greenhouses, and in 4.70% of the rest of greenhouse types. The most widely used heating system is by indirect combustion using heaters equipped with a heat exchanger and a chimney,

followed by direct combustion cannons or heaters with the highest thermal performance. Heating systems using hot water pipes are not used except for specific cases, and many of those installed are not operational because they are not profitable. In general, the use of heating has not been extended due to the very limited days per year that require its use in such a climate. Unlike heating systems, energy-saving systems are used in 44% of greenhouses [43]. To reduce energy waste during the winter period, most use a thermal blanket that is spread over the crop, either directly or on the tutors themselves. The double-wall system, used by 13% of the greenhouses, has a second polyethylene wall inside the greenhouse that allows for better results than those obtained with a simple sheet [48].

### 3.2.6. Energy Supply for Greenhouses

In Almería greenhouses, energy consumption is very little, almost exclusively from irrigation systems, and from the minuscule energy demand of passive climate control systems. Energy consumption represents just over 1.50% of total production costs, nothing comparable, for example, with labor costs, which exceed 45% [9]. This, together with the fact that there is an excellent electricity network in this area, means that projects for the self-consumption of energy from renewable sources are very scarce and generally limited to isolated farms without access to conventional electricity networks. Whereby, the little electricity demand is predominantly supplied from conventional sources of electricity.

The traditional greenhouse climate control in Campo de Dalías do not require significant economic, energy, or water expenditures for their daily operation. According to Valera et al. [43], the energy required to close and open greenhouse windows using electrical drives is only 0.02 MJ of energy per kg of tomatoes produced (for an average production of 19 kg m$^{-2}$), which constitutes a global warming potential (GWP) of 0.003 kg $CO_2$-eq/kg. This represents barely 0.50% of the energy required in a plastic-covered multi-span greenhouse in Almería (4 MJ kg$^{-1}$ and 0.25 kg $CO_2$-eq/kg). In addition, the energy requirements per kilogram of tomatoes produced in a multi-span greenhouse of Almería with natural ventilation are the lowest of all greenhouses worldwide. When the greenhouses from more adverse climatic zones use conventional heating, the associated global energy requirements are on the order from 8 to 32 times greater than those of the non-heated greenhouses of Almería.

### 3.2.7. Cultivation Techniques

Almost 100% of the production in the region is carried out alongside techniques for biological pest control and greenhouse pollination, which results in products containing no chemical residues. Although traditional soil conditioning techniques are still applied by adding organic residual products such as manure, in recent years, the contribution of compost from previous harvest residues that remain in the greenhouse has increased. Soil disinfection is carried out, mainly, with environmentally friendly techniques, such as solarization and biosolarization (combination of solarization with volatile biodegradable substances from organic matter). The passive solarization technique can have an efficiency equal to or greater than that obtained with chemicals such as methyl bromide [49]. The pathogens are eliminated by making use of the heat generated by the sun's energy on the ground covered with a plastic sheet. In greenhouses, solar radiation generates, during solarization, lethal temperatures for most pests and pathogens that inhabit the soil, reaching values higher than 50 °C [50].

## 4. Conclusions

Campo de Dalías is an example of how technology applied to improving agricultural productivity brings about benefits for the environment. Nevertheless, in the historical evolution, the requirements during the first decades of the greenhouses entailed an environmental impact with loss of bush habitat and groundwater overexploitation. Currently, the levels of modernization in greenhouses such as: The automation of drip irrigation, avoiding loss due to evaporation of irrigation ponds, improvement in the construction of

greenhouses, elimination of pesticides, increased biological control of pests, or solarization to avoid pests and pathogens, all favor the system´s sustainability. In relation to the overexploitation of groundwater, the improvement of aquifers implies a diversification of water, promoting desalinated water as well as the reuse of wastewater. This sustainability implies the conservation of existing natural areas with the involvement of all social actors. Therefore, the sustainability of the agricultural production model is conditioned by environmental sustainability, with the modernization of agricultural production playing a fundamental role.

**Author Contributions:** Conceptualization, P.A.A.; methodology, P.A.A., A.P.-F., L.M. and A.J.M.-F.; software, A.J.M.-F.; validation, P.A.A., A.P.-F., L.M. and A.J.M.-F.; formal analysis, P.A.A., A.P.-F. and L.M.; investigation, P.A.A., A.P.-F., L.M. and A.J.M.-F.; resources, P.A.A., A.P.-F., L.M. and A.J.M.-F.; data curation, P.A.A., A.P.-F., L.M. and A.J.M.-F.; writing—original draft preparation, P.A.A., A.P.-F., L.M. and A.J.M.-F.; writing—review and editing, P.A.A., A.P.-F., L.M. and A.J.M.-F.; visualization, P.A.A., A.P.-F., L.M. and A.J.M.-F.; supervision, P.A.A.; project administration, P.A.A.; funding acquisition, P.A.A. All authors have read and agreed to the published version of the manuscript.

**Funding:** Part of this research was funded by the project TRFE-I-2020/001 "BIOSERAREG", through the convocatoria de ayudas a la transferencia de investigacion Transfiere 2020.

**Data Availability Statement:** Data is contained within the article or supplementary material.

**Acknowledgments:** Part of this study has been made possible through the project CEIJ-009 Integrated study of coastal sands vegetation (AREVEG II), sponsored by CEI·MAR.

**Conflicts of Interest:** The authors declare no conflict of interest.

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
