# Peer review of "The Role of Technology in Greenhouse Agriculture: Towards a Sustainable Intensification in Campo de Dalías (Almería, Spain)"

_agronomy, doi:10.3390/agronomy11010101_

Round 1
Reviewer 1 Report
I recommend this paper. It furthers our knowledge of greenhouses and help set guidelines for large scale deployment of greenhouses.
I recommend publication of this paper.
Author Response
Thank you very much for the very favorable comment from the reviewer.
We strongly believe that extending the vision of the modernization of intensive agriculture is essential to preserve a development and operation that is more sustainable and respectful of nature, both in the consumption of resources and in the production of waste.
We will do a review of the manuscript to adjust it and correct the errors of the English.
Reviewer 2 Report
Specific remarks/suggestions.
Line 154- For what reason did the greenhouse areas increase so quickly in the period 1977-1984?
Line 153-199- If available, It would be useful to know also the population and not only the surface settlement
Line 277- In my opinion it is not very clear what “mesh” refers to. Please explain what kind of mesh and, if present, what kind of supports.
Line 315-316- Please, use SI units instead of “gauge”
Line 347- In table2, Dutch Venlo are quite different from “Raspa y amagado”, probably it is a refuse.
Line 400- is it possible for the authors to quantify “mainly”?
General remarks/suggestions
If it is possible, I would suggest to the authors to complete section 3.2 with a paragraph on the energy supply of greenhouses. Only fossil fuel are used? What about renewable energies? Are photovoltaic panels implemented on roofs of greenhouses? In my opinion it is a very important aspect of technology and sustainability.
Author Response
Response to Reviewer 2 Comments
Specific remarks/suggestions.
Point 1: Line 154- For what reason did the greenhouse areas increase so quickly in the period 1977-1984?
Response 1: Done. The following information has been included “The conjunction of several factors such as the low price of agricultural land, the improvement of intensification techniques and the transport of goods to international markets are the main reasons why the greenhouse area increased so rapidly in this period”.
Point 2: Line 153-199- If available, It would be useful to know also the population and not only the surface settlement
Response 2: Done. The following information has been included “in terms of number of inhabitants according to data from the National Institute of Statistics, highlighted Roquetas de Mar (94925), El Ejido (84710); Vícar (25405); Aguadulce (16176)”.
Point 3: Line 277- In my opinion it is not very clear what “mesh” refers to. Please explain what kind of mesh and, if present, what kind of supports.
Response 3: Done. We have changed “mesh” with “coats”.
Point 4: Line 315-316- Please, use SI units instead of “gauge”
Response 4: We understand the comment of the reviewer, however the word “gauge” here refers to the calibre of the plastic.
Point 5: Line 347- In table2, Dutch Venlo are quite different from “Raspa y amagado”, probably it is a refuse.
Response 5: Fixed.
Point 6: Line 400- is it possible for the authors to quantify “mainly”?
Response 6: We have chosen the word "mainly" because this type of technique has become widespread in the area. However, we do not have updated data on its application percentage.
General remarks/suggestions
Point 7: If it is possible, I would suggest to the authors to complete section 3.2 with a paragraph on the energy supply of greenhouses. Only fossil fuel are used? What about renewable energies? Are photovoltaic panels implemented on roofs of greenhouses? In my opinion it is a very important aspect of technology and sustainability.
Response 7: Done
Reviewer 3 Report
- Please extend the introduction section to issues related to the land use changes and groundwater exploitation as the main direct pressures.
- Please correct language and editing errors, g. to describe land use changes and groundwater exploitation as the main direct pressures for change (L. 63-64), 200mm (L. 83) or 5 % (L. 300).
- Please present units in the figures, not in their titles, e.g. Figure 3.
Author Response
Response to Reviewer 3 Comments
Point 1: Please extend the introduction section to issues related to the land use changes and groundwater exploitation as the main direct pressures.
Response 1: Done.
Point 2: Please correct language and editing errors, g. to describe land use changes and groundwater exploitation as the main direct pressures for change (L. 63-64), 200mm (L. 83) or 5 % (L. 300).
Response 2: Fixed.
Point 3: Please present units in the figures, not in their titles, e.g. Figure 3.
Response 3: Fixed.